# SenSE: Semantic-Aware High-Fidelity Universal Speech Enhancement

## Abstract

Generative universal speech enhancement (USE) methods aim to leverage generative models to improve speech quality under various types of distortions. Diffusion- or flow-based generative models are capable of producing enhanced speech with high quality and fidelity. However, they typically achieve speech enhancement by learning an acoustic feature mapping from degraded speech to clean speech, while lacking awareness of high-level semantic information. This deficiency tends to cause semantic ambiguity and acoustic discontinuities in the enhanced speech. In contrast, humans can often comprehend heavily corrupted speech by relying on semantic priors, suggesting that semantics play a crucial role in speech enhancement. Therefore, in this paper, we propose SenSE, which leverages a language model to capture the semantic information of distorted speech and effectively integrates it into a flow-matching-based speech enhancement framework. Specifically, we introduce a semantic-aware speech language model to capture the semantics of degraded speech and generate semantic tokens. We then design a semantic guidance mechanism that incorporates semantic information into the flow-matching-based speech enhancement process, effectively mitigating semantic ambiguity. In addition, we propose a prompt guidance mechanism, which leverages a short reference utterance to alleviate the loss of speaker similarity under severe distortion conditions. The results of several benchmark data sets demonstrate that SenSE not only ensures high perceptual quality but also substantially improves speech fidelity while maintaining strong robustness under severe distortions. Codes and demos are available at [1][2].

## 1 Introduction

Speech enhancement (SE) is designed to improve both the perceptual quality and intelligibility of speech degraded under adverse acoustic conditions, such as background noise (Braun et al., 2021), reverberation (Purushothaman et al., 2023), signal clipping (Mack & Habets, 2019), and bandwidth constraints (Wang & Wang, 2021). Existing speech enhancement methods often focus on handling a single type of distortion, whereas real-world scenarios typically involve diverse and compounded distortions. Therefore, the development of a universal speech enhancement (USE) system capable of addressing arbitrary types of distortion is highly promising for practical applications.

Speech enhancement is fundamentally concerned with two aspects: quality, referring to producing clean and artifact-free signals, and fidelity, referring to preserving the linguistic content and speaker characteristics of the original clean speech. Benefiting from the advances in deep learning, research on speech enhancement has made remarkable progress. Current deep learning-based SE methods can generally be divided into discriminative and generative paradigms. Discriminative approaches (Luo & Mesgarani, 2019; Hu et al., 2020; Hao et al., 2021; Zhao et al., 2022; Fu et al., 2022; Schroter et al., 2022; Rong et al., 2024) typically establish a deterministic mapping from degraded to clean speech. Such approaches have proven effective in suppressing background noise while preserving a relatively high degree of speech fidelity. However, under severe distortions caused by strong broadband noise, such methods often struggle to effectively distinguish speech components from noise, leading to residual artifacts and a perceptible loss of speech content.

---

[1] https://anonymous.4open.science/r/SenSE_ICLR-B262/
[2] https://anonymous.4open.science/w/SenSE-demo-25F9/

Generative models for speech enhancement have recently attracted significant attention. These approaches capture the distribution of clean speech and generate the corresponding clean signal when conditioned on degraded input. They are particularly advantageous for distortions such as clipping or bandwidth limitation, where missing speech components must be reconstructed. Some methods perform speech enhancement by applying language models (LMs) (Wang et al., 2024; Yao et al., 2025; Tang et al., 2025) or masked generative models (MGMs) (Li et al., 2024c; Zhang et al., 2025b) to generate discrete tokens of clean speech produced by a speech codec (Kumar et al., 2023; Li et al., 2024b; Ye et al., 2025a;b; Zhu et al., 2025; Mousavi et al., 2025). These tokens are subsequently decoded by the codec into clean speech. However, since speech codecs typically apply substantial compression, the resulting quantization loss introduces a fundamental bottleneck to achieving high-fidelity speech enhancement. In contrast, diffusion- and flow-based speech enhancement models (Lu et al., 2022; Lemercier et al., 2023; Li et al., 2024a; Scheibler et al., 2024; Lee et al., 2025; Wang et al., 2025a) directly generate continuous acoustic features of clean speech, effectively mitigating the severe quantization loss inherent in discrete tokenization. Nevertheless, these models still operate by mapping distorted acoustic features to clean representations and lack explicit modeling of high-level semantic information. This limitation often leads to semantic ambiguity and acoustic discontinuities under conditions of strong distortion.

Semantics denotes the high-level linguistic meaning in speech, including words, concepts, and contextual cues beyond acoustic features. Recent studies on speech tokenizers (Du et al., 2024) have aimed to encode semantic information into discrete tokens, commonly by clustering representations from self-supervised models (Schneider et al., 2019; Baevski et al., 2020; Hsu et al., 2021; Chen et al., 2022a; 2024) or through text-constrained supervised training for closer alignment with text (Du et al., 2024). While semantic tokens have been widely adopted in text-to-speech (TTS) systems (Borsos et al., 2023; Du et al., 2024) as a natural bridge between text and speech, only a limited number of studies (Yao et al., 2025) have recognized their potential in speech enhancement. In this work, we aim to leverage semantic tokens as an explicit representation of speech semantics to support the enhancement process. However, although semantic tokenizers exhibit inherent robustness to noise, they still produce unreliable representations under heavy noise or severe distortions, which in turn may mislead the speech enhancement model. Language models have been extensively demonstrated to excel at capturing contextual semantics (Lehmann et al., 2023; Jin et al., 2023). Under high noise or strong distortion, crucial speech components are often heavily masked, making it difficult for acoustic feature-based methods to recover them while maintaining semantic coherence. The contextual modeling capability of language models provides a potential solution to this problem.

In this study, we propose SenSE, a novel framework which models the semantics of degraded speech using a language model and integrates this semantic information into flow-matching-based speech enhancement. Specifically, we design a two-stage speech enhancement approach. In the first stage, a semantic-aware speech language model equipped with the $\mathcal{S}^3$ Tokenizer (Du et al., 2024) is used to explicitly model semantic information from degraded speech and generate purified semantic tokens. In the second stage, we introduce a semantic guidance mechanism into flow-matching-based speech enhancement, where both the purified speech semantic tokens and the mel spectrogram of degraded speech are used as conditions to generate the mel spectrogram of enhanced speech. This mechanism enables the model to produce high-quality enhanced speech while maintaining high fidelity. In addition, we propose a prompt guidance mechanism, which allows the model to optionally leverage a short reference utterance from the target speaker to address the loss of speaker similarity that often occurs under severe distortions. Extensive experiments demonstrate that SenSE achieves impressive results in terms of both perceptual quality and fidelity, while maintaining strong robustness under severe distortions.

## 2 Related Work

### 2.1 Generative Speech Enhancement Methods

In recent years, generative model-based approaches for speech enhancement have attracted considerable attention. These methods can be broadly categorized into three paradigms, which respectively leverage LMs, MGMs, and diffusion models or flow matching for speech enhancement. LM-based approaches formulate speech enhancement as a sequence-to-sequence task. SELM (Wang et al.,

2024) applies k-means to quantize noisy speech representations encoded by WavLM (Chen et al., 2022a) into discrete tokens, and then uses a Transformer-based SpeechLM to map noisy tokens to clean tokens. GenSE (Yao et al., 2025) adopts a two-stage design, where an N2S front-end transforms noisy speech into clean semantic tokens, followed by an S2S back-end that generates clean speech from the predicted semantic tokens. LLaSE-G1 (Kang et al., 2025) takes continuous representations encoded by WavLM as input and employs a language model to predict discrete clean-speech tokens. MGM-based approaches iteratively generate discrete speech tokens to reconstruct clean speech. MaskSR (Li et al., 2024c) encodes degraded speech into embeddings that serve as the condition for iteratively generating clean speech tokens. AnyEnhance (Zhang et al., 2025b) builds upon MaskSR by introducing prompt guidance and a self-critic sampling mechanism, further improving model performance. Diffusion- or flow-based approaches directly generate continuous clean speech features conditioned on degraded speech. CDiffSE (Lu et al., 2022) incorporates real noise observations into both the diffusion and reverse diffusion processes, enabling the model to adapt to non-Gaussian noise and generate more natural speech. SGMSE (Richter et al., 2023) extends score-based generative models (SGMs) to the complex STFT domain, employing complex-valued deep networks to directly predict the spectral score distribution. FlowSE (Wang et al., 2025a) conditions on noisy speech mel spectrograms and uses flow matching to generate continuous mel spectrograms of clean speech, achieving high-quality generation with fewer sampling steps. Collectively, these methods demonstrate the strong potential of generative approaches in addressing speech enhancement tasks.

## 2.2 Universal Speech Enhancement

With the rapid progress of speech enhancement, increasing attention has been paid to models that can unify multiple speech enhancement tasks, such as denoising, dereverberation, declipping, and bandwidth extension. Unlike discriminative methods that learn a fixed spectral mapping, generative approaches inherently allow stochasticity, which provides natural advantages for certain distortion types that involve irreversible information loss, such as strong broadband noise, clipping, and band limitation. VoiceFixer (Liu et al., 2022) was the first framework proposed for universal speech enhancement (USE). It employs an analysis module to extract intermediate features from degraded speech, followed by a neural vocoder-based synthesis module to reconstruct waveform. StoRM (Lemercier et al., 2023) adopts a hybrid "predict-and-generate" mechanism, where a predictive model provides a coarse estimate that is subsequently refined through a diffusion-based regeneration process. MaskSR (Li et al., 2024c) utilizes a masked generative modeling (MGM) framework, which progressively generates sequences of clean-speech tokens to restore high-quality speech. AnyEnhance (Zhang et al., 2025b) extends MaskSR by incorporating a prompt-guidance mechanism, further unifying the target speaker extraction (TSE) task. LLaSE-G1 (Kang et al., 2025) employs LLaMA as its backbone, exploring the generalization capability of language models across various speech enhancement tasks. PGUSE (Zhang et al., 2025a) integrates a direct-prediction branch with a diffusion-based branch, effectively combining the strengths of discriminative and generative paradigms to adapt to different tasks. UniFlow (Wang et al., 2025b) leverages flow matching in the latent space to present a unified framework for diverse speech enhancement tasks, exploring the feasibility of consolidating them under a single generative formulation. These studies collectively demonstrate the potential of using a single model to handle a wide range of speech distortions, while also providing important insights and directions for future research in universal speech enhancement.

## 3 Method

### 3.1 Overall Architecture

As illustrated in Fig. 1, SenSE is organized into two key stages: 1) Semantic-Aware Speech Language Model: This module adopts an autoregressive paradigm to derive purified semantic tokens from the continuous speech embeddings obtained by encoding degraded speech. In addition, the $\mathcal{S}^3$ Tokenizer (Du et al., 2024) can optionally be applied to extract semantic tokens from a reference speech signal, providing auxiliary semantic cues. 2) Semantically Guided Speech Enhancement with Flow Matching: In this stage, the purified semantic tokens predicted by SASLM and the reference semantic token (if provided) are combined with their corresponding mel spectrograms and used

as conditioning inputs. A flow-matching-based model then generates the mel spectrogram of clean speech, after which a vocoder reconstructs the enhanced waveform. The details of the Semantic-Aware Speech Language Model are elaborated in Section 3.2, and the Semantic-Guided Speech Enhancement with Flow Matching is described in Section 3.3.

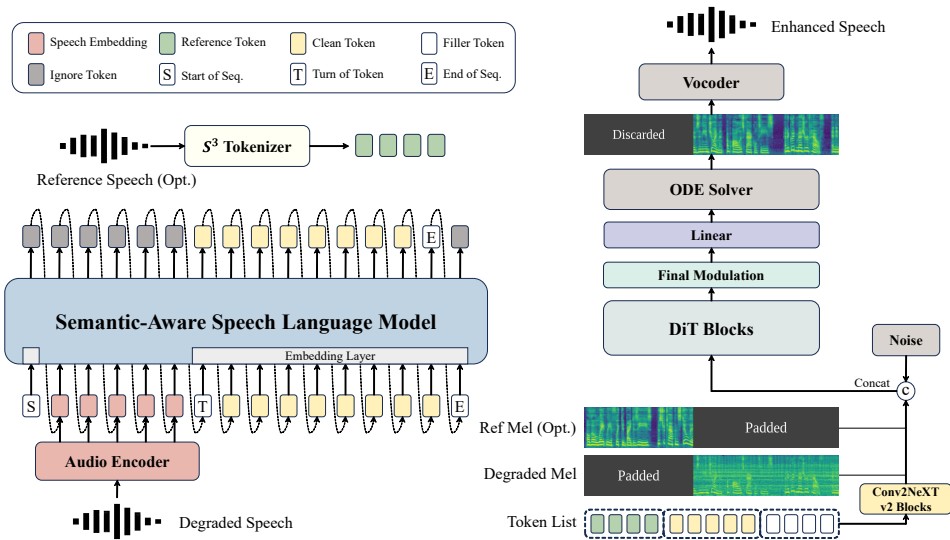

Figure 1: An overview of the two-stage architecture in SenSE with explicit semantic modeling.

## 3.2 SEMANTIC-AWARE SPEECH LANGUAGE MODEL

To enhance robustness under conditions of severe distortion, we introduce a Semantic-Aware Speech Language Model (SASLM). In contrast to the AR-based enhancement framework in GenSE (Yao et al., 2025), our model incorporates an audio encoder that extracts continuous speech embeddings and employs them as input prefixes. By removing the quantization layer typically applied after the encoder, these embeddings preserve richer information, thereby reducing the loss induced by quantization. Furthermore, unlike LLaSE-G1 (Kang et al., 2025), which employs a non-autoregressive feature mapping approach, we adopt a fully autoregressive strategy for semantic token generation. This design effectively exploits the strengths of autoregressive modeling in capturing semantic dependencies and is inherently more amenable to large-scale extension (Henighan et al., 2020). We use a randomly initialized LLaMA model as our backbone, with more detailed configurations provided in Appendix B.1.

We adopt the $\mathcal{S}^3$ Tokenizer as the speech tokenizer. Originally introduced in CosyVoice (Du et al., 2024), it integrates a quantization module, either vector quantization (VQ) (Van Den Oord et al., 2017) or finite scalar quantization (FSQ) (Mentzer et al., 2024), into the encoder of the SenseVoice (An et al., 2024) ASR model to generate discrete tokens under supervised training. By leveraging ASR objectives, the $\mathcal{S}^3$ Tokenizer preserves key semantic information while improving alignment with text, making the tokens well suited for language model-based speech modeling.

For the audio encoder, we initialize with the pretrained Whisper-large-v3[3] (Radford et al., 2023) model and append an adapter composed of fully connected layers. During training, the input sequence to SASLM is formatted as "<Start of Seq. >, speech embedding, <Turn of Token >, speech token, <End of Seq. >", where <Start of Seq. >denotes the start-of-sequence token, <Turn of Token >indicates the transition from speech embeddings to speech tokens, and <End of Seq. >marks the end of the sequence. The model is trained under a next-token prediction objective as shown in Fig. 2a, with the loss computed only over the target token and end-of-sequence segments. The training objective is as follows:

---

[3]https://huggingface.co/openai/whisper-large-v3

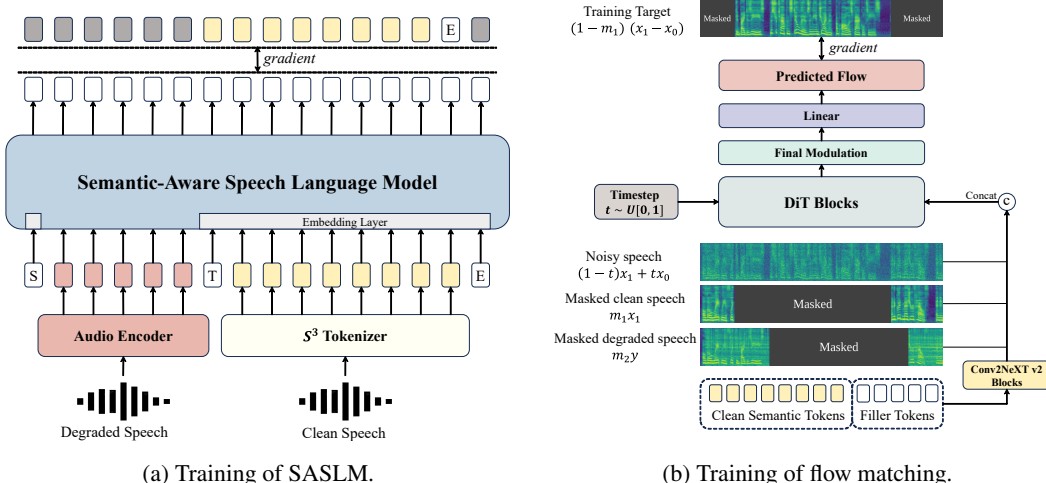

(a) Training of SASLM.

(b) Training of flow matching.

Figure 2: Training pipeline of SenSE. The two stages of the model are trained separately.

$$p(x) = \prod_{k=1}^{n} p\big(s_k \mid s_1, \ldots, s_{k-1}, e_{1\ldots n}\big) \tag{1}$$

where $s$ denotes the semantic tokens of speech, $e$ represents the speech embeddings extracted by the audio encoder, and $n$ indicates the total number of frames in the input speech sequence.

It is worth noting that we first freeze the audio encoder and train only the language model. However, a frozen audio encoder is not fully suitable for the speech enhancement task. Therefore, after training the language model, we freeze it and then perform moderate fine-tuning of the audio encoder, which proves beneficial for improving the overall performance of the model.

At inference time, a sequence consisting of the start-of-sequence symbol, the speech embeddings, and the turn-of-token indicators is provided to the SASLM as a prefix. The model then generates the semantic tokens corresponding to the clean speech via next-token prediction.

### 3.3 SEMANTIC-GUIDED SPEECH ENHANCEMENT WITH FLOW MATCHING

Based on the semantic tokens predicted by SASLM and the reference semantic token (if provided), we incorporate a semantic guidance mechanism into flow-matching-based speech enhancement, which constitutes the second stage of our model. With this mechanism, the model jointly leverages acoustic cues from the degraded mel spectrogram and semantic cues from the semantic tokens. In this way, the model generates clean speech conditioned on semantic tokens, while ensuring that the enhanced output preserves a spectral envelope similar to that of the distorted input, thereby maintaining high fidelity. In addition, we introduce a prompt guidance mechanism at this stage. The model can optionally take a short reference utterance from the target speaker as input, which substantially improves speaker similarity of the enhanced speech. This is particularly beneficial under severe distortion conditions, where the degraded input provides limited information.

**Training** As illustrated in Fig. 2b, we train the second-stage flow matching model on a conditioned speech-infilling task, using a DiT (Peebles & Xie, 2023) backbone. Specifically, we employ DiT blocks following the design of F5-TTS (Chen et al., 2025), in which a zero-initialized adaptive Layer Normalization (adaLN-zero) module is adopted as the final modulation mechanism. The infilling task is defined as generating a missing speech segment based on its surrounding context, incomplete degraded speech, and the complete semantic token sequence. During training, groups of clean speech, degraded speech, and semantic tokens (extracted from the clean speech via the tokenizer) are used as training samples. From the clean and degraded speech, we extract their mel spectrograms. For convenience, we denote the clean speech mel spectrogram as $x_1 \in \mathbb{R}^{F \times T}$, the degraded speech mel spectrogram as $y \in \mathbb{R}^{F \times T}$, and the semantic token sequence as $z$.

Following the flow matching formulation (Lipman et al., 2022), the model input is constructed as $(1-t)x_1 + tx_0$, where $x_0 \sim \mathcal{N}(0, I)$ represents Gaussian noise sampled from the prior distribution, and $t \sim \mathcal{U}[0, 1]$ denotes the sampled flow step. We combine both acoustic cues and semantic cues as the conditioning input to the model. The acoustic cues include the masked clean speech $m_1 \odot x_1$ and the incomplete degraded speech $m_2 \odot y$, where $m_1, m_2$ are binary temporal masks. For the semantic cues, the key challenge lies in aligning the semantic token sequence $z$ with the speech mel spectrogram $x_1, y$, since they differ in temporal length. Inspired by F5-TTS (Chen et al., 2025), semantic tokens are padded with filler tokens to match the length of the mel spectrogram and then encoded by ConvNeXt V2 (Woo et al., 2023) modules to implicitly align the token sequence with the speech features. The condition is constructed by concatenating the clean speech mel-spectrogram, the degraded speech mel spectrogram, and the aligned semantic cues along the feature dimension, based on which the model is trained to reconstruct the masked regions of the clean speech. Therefore, our objective is to train a model $v_\theta$ to predict the velocity $v_\theta((1 - t)x_0 + tx_1, m_1x_1, m_2y, t)$ with target as $v = (1 - m_1)(x_1 - x_0)$. We use the mean squared error (MSE) loss, formally represented as:

$$\mathcal{L}_{\text{FM}} = \mathbb{E}_{t,x_0,x_1} \left\| v_\theta\big((1 - t)x_0 + tx_1, \ m_1x_1, \ m_2y, \ z\big) - (1 - m_1)(x_1 - x_0) \right\|^2 \tag{2}$$

Regarding the rationale for randomly masking the degraded speech, which we refer to as the *degrad mask*, we note that degraded signals often provide direct acoustic cues, which can dominate the learning process. When the model is conditioned simultaneously on degraded speech and semantic tokens, it tends to over-rely on the degraded speech, thereby neglecting the more challenging task of mapping semantic tokens to clean speech. To counteract this bias, we introduce random temporal masking on the degraded speech. This strategy compels the model to reconstruct the masked segments by exploiting semantic tokens together with the corresponding masked regions of the clean speech. Through this masking-based training paradigm, the model learns to align semantic tokens with both clean and degraded speech. In particular, the model extracts as much spectral envelope information as possible from the degraded signal and integrates it with the higher-level semantic content captured by the tokens to reconstruct clean speech.

**Inference** Based on the flow matching training strategy described above, the model learns to implicitly align semantic tokens with acoustic conditions and effectively integrate both sources of information to generate clean speech. We thus design the following inference procedure.

To generate enhanced speech, we prepare the degraded speech $y_{degraded} \in \mathbb{R}^{F \times T_1}$, the semantic tokens $z_{gen}$ predicted by SASLM , and an optional reference speech $x_{ref} \in \mathbb{R}^{F \times T_2}$. During inference, as shown in Fig. 1, we pad the tail of $x_{ref}$ with empty frames of length $T_1$, which serves as the portion to be generated. Similarly, we pad the front of $y_{degraded}$ with empty frames of length $T_2$ so that it aligns with the generative portion, and the semantic token sequence $z_{gen}$, formed by concatenating the reference tokens and the purified tokens, is extended with filler tokens in the same way as in training. When no reference speech is provided, $T_2$ is set to zero, resulting in an all-empty reference channel without requiring additional padding for $y_{degraded}$. The model estimates the velocity field $v$, and an ODE solver is applied to generate the enhanced mel spectrogram. Finally, the mel spectrogram is converted into a waveform using a pretrained BigVGAN vocoder[4] (gil Lee et al., 2023).

## 4 EXPERIMENT AND RESULTS

### 4.1 EXPERIMENTAL SETUP

**Datasets** We use clean speech data from the open-source Emilia (He et al., 2024) dataset to train our base model. We selected approximately 22,000 hours of data with DNSMOS scores greater than 3.40. Additionally, small models were trained for ablation studies using 1,400 hours of clean speech from the Interspeech 2020 DNS Challenge (Reddy et al., 2020) dataset. The noise datasets included DEMAND, ESC-50, and DNS Challenge (Cutler et al., 2024), totaling roughly 700 hours, while room impulse responses (RIRs) were sourced from openSLR26 and openSLR28 (Ko et al., 2017). For training the SASLM, all samples were resampled to 16 kHz, whereas the flow matching

---

[4]https://github.com/NVIDIA/BigVGAN

Table 1: Results on DNS Challenge *no-reverb* and DNS Challenge HardSet for denoising. The boldface indicates the best result and the underline denotes the second best. 'D' and 'G' denote discriminative and generative methods, respectively.

| Model | Type | DNSMOS↑ | NISQA↑ | Speech-BERTScore↑ | dWER(%)↓ | SIM-o↑ |
|---|---|---|---|---|---|---|
| **DNS Challenge *no-reverb*** | | | | | | |
| Voicefixer | D | 3.248 | 4.385 | 0.861 | 9.28 | 0.714 |
| TF-GridNet | D | 3.312 | 4.332 | 0.930 | 4.22 | **0.956** |
| MP-SENet | D | 3.271 | 4.339 | 0.904 | 5.68 | 0.944 |
| CleanMel | D | 3.375 | 4.346 | 0.934 | 4.21 | 0.912 |
| PGUSE | D+G | 3.333 | 4.661 | 0.932 | 4.40 | 0.942 |
| SELM | G | 3.258 | 3.802 | 0.762 | 44.42 | 0.211 |
| GenSE | G | **3.425** | 4.672 | 0.838 | 20.08 | 0.285 |
| FlowSE | G | 3.265 | 4.733 | 0.898 | 9.92 | 0.846 |
| LLaSE-G1 | G | 3.415 | 4.504 | 0.889 | 8.69 | 0.748 |
| AnyEnhance | G | 3.406 | 4.784 | 0.925 | 4.95 | 0.885 |
| SenSE | G | 3.376 | **4.788** | **0.942** | **3.89** | 0.921 |
| **DNS Challenge HardSet** | | | | | | |
| Voicefixer | D | 3.119 | 3.958 | 0.779 | 27.01 | 0.553 |
| TF-GridNet | D | 3.146 | 3.974 | 0.844 | 12.00 | 0.813 |
| MP-SENet | D | 3.074 | 4.037 | 0.793 | 23.49 | 0.815 |
| CleanMel | D | 3.339 | 3.695 | 0.882 | 9.90 | 0.803 |
| PGUSE | D+G | 3.251 | 4.112 | 0.871 | 14.50 | **0.843** |
| FlowSE | G | 2.940 | 4.057 | 0.799 | 29.38 | 0.684 |
| LLaSE-G1 | G | 3.370 | 4.366 | 0.830 | 28.47 | 0.604 |
| AnyEnhance | G | 3.384 | 4.817 | 0.876 | 15.48 | 0.778 |
| SenSE | G | **3.408** | **4.873** | **0.915** | **8.93** | 0.838 |

stage was trained on 24 kHz audio, with all other sampling rates resampled to 24 kHz. Based on the above corpora, we construct paired training data by applying a distortion simulation pipeline to clean speech. Further details of data simulation are provided in Appendix B.2.

For the speech denoising task, we evaluate on the official DNS Challenge no-reverb test set. In addition, to assess model performance under low signal-to-noise ratio (SNR) conditions, we construct a new test set, DNS Challenge HardSet, by re-mixing the speech and noise from the DNS Challenge no-reverb test set with SNR randomly sampled from the range (-5, 0) dB. To account for multiple distortion types, we further evaluate on the simulated DNS Challenge GSR and VCTK GSR test sets. In DNS Challenge GSR, clean speech is processed with our simulation pipeline to introduce distortions and then re-mixed with noise. In VCTK GSR, clean speech from the VCTK dataset is similarly processed using our simulation pipeline and mixed with unseen noise and RIRs, resulting in a total of 167 samples.

**Comparison Models**   We compare our model against state-of-the-art speech enhancement systems, including both discriminative models: VoiceFixer (Liu et al., 2022), TF-GridNet (Wang et al., 2023), MP-SENet (Lu et al., 2025), and CleanMel (Shao et al., 2025); as well as generative models: SELM (Wang et al., 2024), GenSE (Yao et al., 2025), AnyEnhance (Zhang et al., 2025b), LLaSE-G1 (Kang et al., 2025), and FlowSE (Wang et al., 2025a); as well as the hybrid predictive-generative model PGUSE (Zhang et al., 2025a). Details of baseline models see Appendix A.

**Metrics**   We evaluate our model using several commonly adopted metrics for speech enhancement. **DNSMOS:** a reference-free perceptual quality metric (Reddy et al., 2021). **NISQA:** a reference-free perceptual quality metric designed for 48 kHz speech (Mittag et al., 2021). **Speech-BERTScore:** a metric for assessing the semantic similarity between enhanced and reference speech (Saeki et al., 2024). **dWER:** a metric that evaluates the word error rate (WER) difference between enhanced and reference speech using ASR transcriptions. **SIM-o:** a measure of timbre similarity between enhanced and reference speech (Le et al., 2023). **PESQ:** the wideband Perceptual Evalua-

Table 2: Results on the DNS Challenge GSR and VCTK GSR simulated test sets, which cover diverse distortion types including noise, reverberation, clipping, and band limitation. The boldface indicates the best result, and the underline denotes the second best. 'D' and 'G' denote discriminative and generative methods, respectively.

| Model | Type | DNSMOS↑ | NISQA↓ | Speech-BERTScore↑ | dWER(%)↓ | SIM-o↑ |
|---|---|---|---|---|---|---|
| **DNS Challenge GSR** | | | | | | |
| Voicefixer | D | 3.285 | 4.003 | 0.817 | 18.13 | 0.553 |
| TF-GridNet | D | 3.233 | 4.041 | 0.872 | 7.36 | 0.767 |
| PGUSE | D+G | 3.291 | 4.166 | 0.896 | 8.52 | **0.805** |
| AnyEnhance | G | **3.396** | **4.847** | 0.896 | 12.25 | 0.779 |
| SenSE | G | 3.388 | 4.806 | **0.922** | **6.70** | 0.797 |
| **VCTK GSR** | | | | | | |
| Voicefixer | D | 2.976 | 4.009 | 0.887 | 8.30 | 0.681 |
| TF-GridNet | D | 3.054 | 4.425 | 0.924 | 3.77 | 0.855 |
| PGUSE | D+G | 3.057 | 4.419 | 0.928 | 3.10 | **0.879** |
| AnyEnhance | G | **3.128** | **4.756** | 0.924 | 4.62 | 0.795 |
| SenSE | G | 3.109 | 4.726 | **0.943** | **1.79** | 0.824 |

Table 3: Effect of prompt guidance mechanism.

| Model | DNSMOS↑ | PESQ↑ | ESTOI↑ | Speech-BERTScore↑ | dWER(%)↓ | SIM-o↑ |
|---|---|---|---|---|---|---|
| **DNS Challenge-Hardset** | | | | | | |
| SenSE | **3.408** | 2.071 | 0.854 | 0.916 | **8.91** | 0.837 |
| SenSE (w/ prompt) | **3.408** | **2.218** | **0.859** | **0.920** | 9.30 | **0.883** |
| **DNS Challenge GSR** | | | | | | |
| SenSE | **3.389** | 2.219 | 0.865 | 0.922 | **6.73** | 0.799 |
| SenSE (w/ prompt) | 3.377 | **2.416** | **0.873** | **0.928** | 7.05 | **0.874** |

tion of Speech Quality (Rix et al., 2001). **ESTOI:** the Extended Short-Time Objective Intelligibility measure (Jensen & Taal, 2016). Further details of these metrics can be found in Appendix B.3.

## 4.2 EXPERIMENTAL RESULTS

Tab. 1 and 2 present the main results on speech denoising and the general speech restoration tasks. We categorize all methods into two types: discriminative and generative. In the Type column of the table, we use "D" to denote discriminative methods and "G" to denote generative methods. For PGUSE, which combines both paradigms, we classify it as a discriminative method since the discriminative component plays the dominant role. During inference of SenSE, sampling in SASLM is disabled to ensure relatively stable outputs. We report the average score of SenSE and the open-sourced sampling-based baselines using five different random seeds. By default, we set the CFG strength (Ho & Salimans, 2022) to 0.5 and the sway sampling coefficient (Chen et al., 2025) to -1 for our flow matching model. Note that all enhanced outputs from the models are resampled to 16 kHz.

Based on the results on the DNS Challenge no-reverb test set, we make the following observations: 1) For perceptual quality metrics (DNSMOS and NISQA), SenSE achieves competitive or superior results. Specifically, SenSE obtains a DNSMOS of 3.374 and a NISQA of 4.788. The DNSMOS score is comparable to the best baseline, GenSE (3.425), while the NISQA score surpasses all baselines. 2) For semantic-related metrics (SpeechBERTScore and dWER), SenSE outperforms both discriminative and generative baselines, achieving the best results. It is noteworthy that generative approaches typically do not perform as well as discriminative ones on dWER; thus, SenSE highlights exceptional semantic fidelity. 3) For the speaker similarity metric (SIM-o), SenSE achieves a score of 0.920, which significantly surpasses all generative baselines and is very close to the best discriminative baseline, TF-GridNet (0.956).

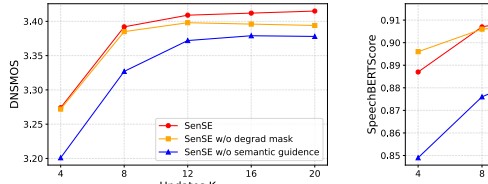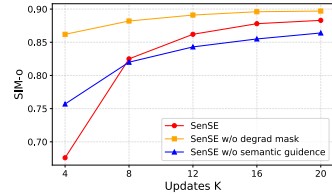

Figure 3: Results of the ablation study on the SenSE framework, where "w/o" indicates the removal of a specific method or component.

On the more challenging DNS Challenge HardSet, the advantages of SenSE become more pronounced, demonstrating strong robustness under low-SNR conditions. SenSE achieves the best results on DNSMOS, NISQA, SpeechBERTScore, and dWER, with overall performance substantially surpassing all baselines. For SIM-o, SenSE obtains a score of 0.837, only slightly lower than the best result from PGUSE (0.842).

On the DNS Challenge GSR and VCTK GSR test sets, we evaluate SenSE's ability to handle diverse distortion types. Across both datasets, SenSE achieves DNSMOS and NISQA scores comparable to AnyEnhance, while significantly outperforming discriminative baselines. At the same time, SenSE substantially surpasses all baselines on semantic metrics (SpeechBERTScore and dWER), demonstrating superior semantic fidelity. For SIM-o, SenSE achieves higher scores than AnyEnhance and reaches performance comparable to discriminative baselines. The evaluation of PESQ and STOI metrics on the test set is presented in Appendix B.8, and the spectrogram visualizations of the enhancement results for each model are also provided in Appendix B.9.

We further evaluate the effect of the prompt guidance mechanism on two challenging test sets, the DNS Challenge HardSet and the DNS Challenge GSR. For each test utterance (10 seconds in length), we randomly sample a 3-second segment from the corresponding clean speech as the prompt. As shown in Tab. 3, the introduction of the prompt yields substantial improvements on reference-based metrics (PESQ, ESTOI, SpeechBERTScore, SIM), with particularly notable gains in speaker similarity, while causing only minor performance degradation on DNSMOS and dWER.

## 4.3 ABLATION STUDIES AND ANALYSIS

To validate the effectiveness of the proposed degrad mask and semantic guidance mechanism in SenSE, we conduct ablation experiments. We trained the two components of the small model for 200k updates using the DNS Challenge 1,400h clean speech dataset. Further details on the model configuration are provided in Appendix B.1. Specifically, 1) we removed the mask applied to degraded speech in the flow matching stage; 2) we removed the first-stage SASLM and eliminated the semantic token from the conditions in the second-stage flow matching module.

As shown in Fig. 3, removing the degrad mask yields lower DNSMOS scores compared to the full SenSE framework, indicating a decline in perceptual speech quality. For SpeechBERTScore, the degrad mask also provides a positive effect. This supports our insight that applying a mask to the degraded speech encourages the enhancement model to rely more on semantic tokens, thereby improving semantic accuracy. Although the degrad mask initially reduces speaker similarity, the results in the figure show that this gap gradually narrows as training progresses. Overall, the degrad mask mechanism enables SenSE to achieve superior performance across all evaluation metrics.

For DNSMOS and SpeechBERTScore, the semantic guidance mechanism yields substantial performance improvements, indicating that semantic conditioning effectively mitigates the semantic ambiguity inherent in flow-matching-based speech enhancement methods and, to some extent, enhances acoustic continuity. Regarding speaker similarity, the semantic guidance mechanism initially leads to lower similarity at early training stages; however, as training progresses, models equipped with semantic guidance exhibit a marked improvement, ultimately surpassing those without it. These findings demonstrate that semantic guidance not only reduces ambiguity at the semantic level but also contributes to improved speech quality and speaker similarity by reinforcing semantic clarity.

## 5 CONCLUSION

In this study, we propose SenSE, a semantics-aware universal speech enhancement framework designed for high fidelity. It consists of two key components: 1) a semantic-aware speech language model, which extracts high-level semantic information from degraded speech to generate purified semantic tokens; 2) a semantic-guided speech enhancement module with flow matching, which integrates semantic knowledge from purified semantic tokens into a flow-matching-based speech enhancement method via a semantic guidance mechanism, thereby substantially improving its semantic representation capability while maintaining high fidelity. In addition, we introduce a prompt guidance mechanism to address the loss of speaker similarity caused by severe distortions. We will release our code and models to facilitate reproducibility and further research in this field.

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

# A    BASELINE DETAILS

**VoiceFixer** (Liu et al., 2022)    A speech restoration system that unifies denoising, dereverberation, and bandwidth extension in a single framework. It employs a two-stage end-to-end architecture: in the Analysis stage, a discriminative method is first applied to the mel-spectrogram to restore the distorted speech, followed by the Synthesis stage, where a vocoder reconstructs clean speech from the enhanced mel spectrogram. We use the official code and pre-trained checkpoint to evaluate[5].

**TF-GridNet** (Wang et al., 2023)    A time-frequency domain speech enhancement network built on a grid-structured dual-path architecture. It performs iterative estimation in both temporal and frequency dimensions for superior separation quality. We compare with our reproduced 8.5M TF-GridNet on approximately 1,400 hours of clean speech from the DNS Challenge and the same noise data used in our paper.

**SELM** (Wang et al., 2024)    A speech enhancement model that transforms raw noisy waveform into discrete tokens using a pretrained self-supervised learning model and k-means tokenizer, uses a language model to predict clean tokens, then detokenizes via HiFi-GAN  (Kong et al., 2020) to produce enhanced speech. We use the official inference results on DNS Challenge blind test set provided by the authors for evaluation.

**CleanMel** (Shao et al., 2025)    A single-channel mel spectrogram denoising and dereverberation model that aims to improve both speech quality and ASR accuracy. It maps noisy and reverberant mic recordings to clean mel spectrograms, which can either be converted to waveform via a neural vocoder or fed directly into ASR. We use the official code and pre-trained checkpoint to evaluate[6].

**GenSE** (Yao et al., 2025)    A generative speech enhancement system that treats SE as a conditional language modeling task rather than pure continuous-signal regression. GenSE uses a hierarchical modeling approach: first a Noisy-to-Semantic (N2S) module that denoises in the semantic token space, then a Semantic-to-Speech (S2S) module that generates clean acoustic tokens using a token-chain prompting mechanism. We use the official inference results on DNS Challenge blind test set provided by the authors for evaluation.

**AnyEnhance** (Zhang et al., 2025b)    A unified generative model for voice enhancement that handles speech and singing, supporting multiple tasks (denoising, dereverberation, declipping, super-resolution, target speaker extraction) without fine-tuning via a masked generative architecture. We compare with our reproduced 323M AnyEnhance on the same training data of our base model.

**LLaSE-G1** (Kang et al., 2025)    A LLaMA-based speech enhancement model designed for strong generalization across diverse SE tasks by using continuous features from WavLM  (Chen et al., 2022a) as input and predicting discrete tokens from X-Codec2 to better preserve acoustic fidelity. It supports multiple tasks under a dual-channel input/output setup without needing task-specific IDs. We use the official code and pre-trained checkpoint to evaluate[7].

**FlowSE** (Wang et al., 2025a)    A speech enhancement model based on conditional flow matching that learns a continuous one-shot transformation from noisy to clean mel spectrograms, greatly reducing inference latency compared to diffusion-based methods. We use the official code and pretrained checkpoint to evaluate[8].

**PGUSE** (Zhang et al., 2025a)    A universal speech enhancement model combining predictive and generative modeling: one branch directly predicts clean speech from degraded input, the other uses a diffusion-based generative branch. The two are fused via output fusion and a truncated diffusion scheme. We use the official code[9] and retrained on approximately 1,400 hours of clean speech from the DNS Challenge and the same noise data used in our paper.

---

[5]https://github.com/haoheliu/voicefixer
[6]https://github.com/Audio-WestlakeU/CleanMel
[7]https://github.com/ASLP-lab/LLaSE-G1
[8]https://github.com/Honee-W/FlowSE
[9]https://github.com/hyyan2k/PGUSE

Table 4: Constructed Distortion Types and Probabilities in Training Data Simulation.

| Family | Probability | Hyperparameters |
|---|---|---|
| Noise | 0.9 | $SNR \in [-10, 10]$ |
| Reverberation | 0.5 | - |
| Clipping | 0.25 | Clipping threshold $\in [0.05, 0.90]$ |
| Bandwidth Limitation | 0.5 | Bandwidth $\in \{2, 4, 8, 16, 22.05\} kHz$ |

## B  EXPERIMENTAL RESULT SUPPLEMENTS

### B.1  MODEL CONFIGURATION

For the Base model, the SASLM was trained on 8 NVIDIA RTX 5880 Ada Generation GPUs with a batch size of 76,800 audio frames for 1.15M steps. During the first 550K steps, the audio encoder was frozen while only the language model was optimized; in the subsequent steps, the language model was frozen, and the audio encoder was fine-tuned. The flow matching stage was trained on the same GPUs with a batch size of 153,600 audio frames for 350K steps. For the Small model, both stages were trained on 8 NVIDIA RTX 4090 GPUs with a batch size of 147,200 audio frames for 200K steps. Across all training configurations, the AdamW optimizer was employed with a peak learning rate of 7.5e-5, using 20k linear warm-up steps followed by linear decay.

The detailed configurations of the Base and Small models are as follows:

- **Base model**: Whisper is adopted as the audio encoder with an output dimension of 1280; the SASLM Transformer is configured with a hidden size of 1536, 20 layers, and 16 attention heads; the DiT Blocks Transformer is configured with a hidden size of 1024, 22 layers, and 16 attention heads; ConvNeXt V2 is configured with 512 embedding dimensions and 1024 FFN dimensions.

- **Small model**: Conformer is adopted as the audio encoder with an output dimension of 512; the SASLM Transformer is configured with a hidden size of 1024, 6 layers, and 16 attention heads; the DiT Blocks Transformer is configured with a hidden size of 768, 18 layers, and 12 attention heads; ConvNeXt V2 is configured with 512 embedding dimensions and 1024 FFN dimensions.

During training, the masking ratio for clean speech mel frames was set between 70% and 100%, whereas the ratio for degraded speech mel frames ranged from 50% to 100%. In addition, the mel spectrograms of clean speech were subjected to dropout with a probability of 50%, and clean speech, degraded speech, and semantic tokens were further dropped out with an additional probability of 20%. These stochastic perturbations were introduced to support CFG training and to reduce the model's over-reliance on clean speech conditions during optimization.

### B.2  DETAILS OF DATA SIMULATION

The task of universal speech enhancement is typically addressed by constructing simulated data from three independent sources: a clean speech dataset, a noise dataset, and a RIR dataset. While the main text provides an overview of the dataset composition, this section presents a detailed description of the data simulation process.

We constructed paired training data through the following simulation pipeline. First, clean speech, noise, and RIR were randomly sampled. Reverberation was simulated by convolving clean speech with an RIR with a probability of 50%. Clipping distortion was applied by thresholding the waveform at a randomly selected value between 0.05 and 0.9, with a 25% probability. To emulate bandwidth limitation, speech was randomly downsampled (50% probability) to one of five sampling rates: 2 kHz, 4 kHz, 8 kHz, 16 kHz, or 22.05 kHz. Finally, additive noise was introduced with a 50% probability, and the SNR was uniformly sampled from [-10, 10] dB. The simulation parameters are summarized more intuitively in Tab. 4.

### B.3 DETAILED INTRODUCTION OF EVALUATION METRICS

We employ a comprehensive suite of evaluation metrics to rigorously assess the proposed model, including perceptual quality, semantic similarity, speaker consistency, and signal-level fidelity.

**DNSMOS** (Reddy et al., 2021)   A non-intrusive speech quality assessment metric proposed by Microsoft as part of the Deep Noise Suppression (DNS) Challenge. It is designed to predict human Mean Opinion Scores (MOS) for speech quality without requiring a clean reference signal. It provides three scores ranging from 1 to 5: SIG (signal quality), BAK (background noise quality), and OVRL (overall quality). In this work, only the OVRL score is used as the primary indicator of perceptual quality.

**NISQA** (Mittag et al., 2021)   A reference-free speech quality evaluation metric. It aims to predict human Mean Opinion Scores (MOS) without requiring a clean reference signal, making it suitable for speech communication, enhancement, and real-time noise suppression scenarios. It is designed for 48 kHz speech and outputs a score between 1 and 5 to evaluate overall perceptual quality.

**SpeechBERTScore** (Saeki et al., 2024)   A reference-aware automatic evaluation method for speech generation, inspired by NLP evaluation metrics such as BERTScore. It is designed to measure similarity between generated speech and reference speech in semantic/feature space, aiming to provide a metric that correlates well with human subjective judgments. In this work, we use HuBERT-base[10] model to extract the feature.

**dWER** (Wang et al., 2021)   A metric that evaluates semantic preservation by measuring the difference between the WERs of synthesized and reference transcriptions obtained from an ASR system. We adopt Whisper-large-v3 (Radford et al., 2023) as the ASR model for this evaluation.

**SIM-o** (Le et al., 2023)   An objective speaker similarity metric that quantifies how closely a generated or enhanced speech sample preserves the speaker identity of a reference recording. In this work, embeddings are extracted using a WavLM-large-based (Chen et al., 2022b) speaker verification model, and cosine similarity is computed between synthesized and ground-truth speech signals.

**PESQ** (Rix et al., 2001)   A standardized objective metric that estimates subjective mean opinion scores (MOS) for normal-hearing listeners. Originally designed to assess audio quality in noisy or distorted telephone networks, it has become a widely adopted measure for evaluating the clarity and overall quality of speech enhancement algorithms. PESQ scores typically range from 1.0 to 4.5.

**STOI** (Taal et al., 2011)   A metric which objectively measures speech intelligibility, ranging from 0 to 1. It is especially effective for assessing speech in conditions with temporally modulated noise or time-frequency processing, and is intended for normal-hearing listeners.

### B.4 COMPARISON OF SPEECH TOKENIZERS

In our experiments, we adopt $\mathcal{S}^3$ Tokenizer v1 50hz as the speech tokenizer. CosyVoice provides three different versions of the tokenizer, namely $\mathcal{S}^3$ Tokenizer v1 50hz, $\mathcal{S}^3$ Tokenizer v1 25hz, and $\mathcal{S}^3$ Tokenizer v2 25hz. The difference between v1 50hz and v1 25hz lies only in the frame rate, whereas v1 and v2 employ different tokenizer architectures. To examine the impact of different tokenizers, we trained three corresponding versions of our model using these tokenizers.

The results in Tab. 5 show that using $\mathcal{S}^3$ Tokenizer v1 50hz yields better overall performance. Notably, the v1 50hz tokenizer achieves higher perceptual scores and SpeechBERTScore, whereas the v1 25hz tokenizer produces better dWER and SIM results. We attribute this to the fact that tokens at 50 Hz contain richer information, which benefits the flow matching process in generating higher-quality speech, thereby improving perceptual metrics. In contrast, the lower frame rate of 25 Hz tokens is more favorable for language model training, leading to better performance on dWER.

---

[10]https://huggingface.co/facebook/hubert-base-ls960.

Table 5: Results of the comparison study on the speech tokenizer.

| Tokenizer | DNSMOS↑ | NISQA↑ | Speech-BERTScore↑ | dWER(%)↓ | SIM-o↑ |
|---|---|---|---|---|---|
| $\mathcal{S}^3$ Tokenizer v1 50hz | **3.407** | **4.837** | **0.914** | 10.19 | 0.873 |
| $\mathcal{S}^3$ Tokenizer v1 25hz | 3.369 | 4.640 | 0.912 | **9.91** | **0.883** |
| $\mathcal{S}^3$ Tokenizer v2 25hz | 3.400 | 4.788 | 0.914 | 11.15 | 0.877 |

Table 6: Results of the comparison study on the audio encoder.

| Audio Encoder | DNSMOS↑ | NISQA↑ | Speech-BERTScore↑ | dWER(%)↓ | SIM-o↑ |
|---|---|---|---|---|---|
| Conformer | **3.407** | 4.837 | 0.914 | 10.19 | 0.873 |
| Whisper | 3.396 | 4.860 | **0.930** | **5.25** | **0.877** |
| WavLM | 3.394 | **4.863** | 0.926 | 5.63 | 0.874 |

## B.5 COMPARISON OF AUDIO ENCODERS

To investigate the impact of different audio encoders on the model, we performed a comparative study of three encoders: Conformer, WavLM (large), and Whisper (large-v3). The Conformer encoder was randomly initialized, whereas WavLM and Whisper were initialized with publicly available pretrained weights. During training, the Conformer was updated jointly with the language model, while WavLM and Whisper remained frozen.

The experimental results are presented in Tab. 6. As shown, using pretrained audio encoders yields better performance than the randomly initialized Conformer, and the Whisper encoder achieves overall stronger results compared with WavLM. This indicates that the prior knowledge embedded in pretrained audio encoders benefits the language model in capturing semantic information from degraded speech. However, most existing audio encoders are trained primarily for speech recognition or speech understanding tasks, and therefore have limited ability to encode distorted speech. As a result, these encoders may not be ideally suited for speech enhancement. To mitigate this limitation, we adopt a fine-tuning strategy for the audio encoder, which helps adapt it more effectively to the enhancement task.

## B.6 COMPARISON OF VOCODERS

We evaluate the impact of different vocoders on the performance of SenSE for speech enhancement, focusing on a comparison between BigVGAN (gil Lee et al., 2023) and Vocos (Siuzdak, 2023), both of which are tested using their publicly available 24 kHz pretrained weights.

As shown in Tab. 7, the experimental results demonstrate that using BigVGAN as the vocoder yields overall better performance compared with Vocos, with the only exception being a slightly lower score on the dWER metric.

## B.7 ANALYSIS OF INFERENCE-TIME PARAMETERS

In this section, we conduct a comparative study on various inference parameters of flow matching, with results shown in Fig. 4. Our findings can be summarized as follows: 1) Number of Function Evaluations (NFE): The model achieves its best performance when NFE is set to 8. Increasing the number of steps beyond this point leads to a decline in performance. 2) Classifier-Free Guidance (CFG): A moderate level of CFG improves overall enhancement quality. Specifically, a CFG strength of 0.5 yields higher SpeechBERTScore and relatively better DNSMOS. However, larger CFG values significantly degrade performance. 3) Sway Sampling: This strategy substantially improves enhancement outcomes, indicating that allocating more inference steps to the early phase of the process benefits the model and leads to better results.

Table 7: Results of the comparison study on the vocoder.

| Vocoder | DNSMOS↑ | NISQA↑ | Speech-BERTScore↑ | dWER(%)↓ | SIM-o↑ |
|---------|---------|--------|-------------------|----------|--------|
| BigVGAN | **3.407** | **4.837** | **0.914** | 10.19 | **0.873** |
| Vocos | 3.396 | 4.795 | 0.914 | **8.63** | 0.861 |

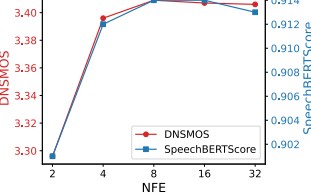 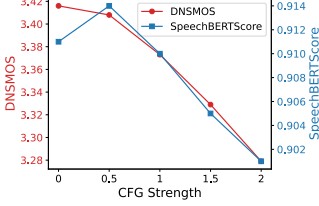 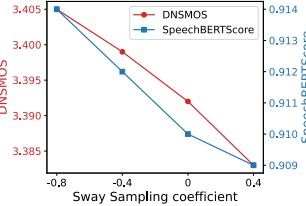

Figure 4: Analysis of the three inference-time parameters: NFE, CFG strength, and the sway sampling coefficient.

### B.8 EVALUATION ON PESQ AND ESTOI

Many speech enhancement studies (Wang et al., 2024; Yao et al., 2025; Zhang et al., 2025b) have demonstrated that generative methods face an inherent disadvantage in PESQ and ESTOI evaluation compared with discriminative models, primarily because they do not reconstruct speech at the waveform level. This limitation stems from the fact that generative models tend to discard acoustical details that are perceptually less salient. Nevertheless, these metrics remain applicable for evaluating the extent to which generative models preserve the essential information of the original speech. Accordingly, we evaluate SenSE and other purely generative methods using PESQ and ESTOI on the DNS Challenge no-reverb, DNS Challenge HardSet, and DNS Challenge GSR datasets.

As shown in Tab. 8, SenSE achieves higher PESQ and ESTOI scores than all baselines across all test sets, further demonstrating its fidelity advantage over other generative models.

### B.9 VISUALIZE OF THE ENHANCED SPECTROGRAM

In this section, we visualize the enhanced spectrograms produced by SenSE and selected baseline models to provide a more intuitive understanding of the advantages of the proposed approach. For evaluation, we select a challenging sample from the DNS Challenge containing strong broadband noise. We present the spectrograms of the noisy input, the outputs of different enhancement models, and the clean reference, along with their corresponding transcriptions obtained using the Whisper-large-v3 speech recognition model.

As shown in Fig. 5, TF-GridNet, representing discriminative models, loses a considerable amount of speech detail compared with the clean target, particularly transient consonants and high-frequency components. PGUSE, which combines discriminative and generative models, alleviates the high-frequency loss to some extent, yet still suffers from substantial detail degradation. AnyEnhance, as a representative generative model, effectively preserves fine-grained details but introduces noticeable semantic deviations from the clean target. In contrast, the proposed SenSE outperforms all baselines by better preserving both speech details and semantic consistency, further demonstrating its effectiveness.

### B.10 LIMITATIONS AND FUTURE WORKS

We identify two main limitations of SenSE.

**Training data limitations.** For training, we leverage nearly 22,000 hours of clean speech from the Emilia dataset, which is significantly larger than what has typically been used in prior speech enhancement studies. However, due to resource constraints, only 700 hours of noise data and a limited amount of RIR data are included. Previous research (Zhang et al., 2025c) has shown that

Table 8: PESQ/ESTOI on DNS Challenge test sets.

| Model | DNS Challenge no-reverb | | DNS Challenge HardSet | | DNS Challenge GSR | |
|---|---|---|---|---|---|---|
| | PESQ | ESTOI | PESQ | ESTOI | PESQ | ESTOI |
| SELM | 1.075 | 0.463 | – | – | – | – |
| GenSE | 1.130 | 0.592 | – | – | – | – |
| FlowSE | 2.150 | 0.859 | 1.377 | 0.698 | – | – |
| LLaSE-G1 | 1.440 | 0.636 | 1.440 | 0.636 | – | – |
| AnyEnhance | 2.561 | 0.884 | 1.814 | 0.809 | 2.088 | 0.841 |
| SenSE | **2.799** | **0.924** | **2.070** | **0.854** | **2.219** | **0.865** |

the diversity of noise types strongly influences SE performance. Hence, the limited noise and RIR coverage may restrict the full potential of SenSE. Moreover, language models are generally believed to benefit greatly from scaling, where larger model sizes and richer datasets can yield substantial performance gains. In this regard, the availability of training data and computational resources may also impose an upper limit on the performance of SenSE.

**Model size and computational cost.** Although SenSE achieves remarkable performance, its complex architecture incurs a large number of parameters and considerable computational overhead. This stems from our focus in this work on exploring the upper bound of speech enhancement performance. Nevertheless, we believe that every component in the SenSE pipeline can be further optimized in terms of parameter count and computational complexity, without substantially sacrificing performance. For instance, recent work such as MeanFlow (Geng et al., 2025) has explored one-step generative modeling, offering valuable insights into improving the efficiency of flow matching approaches. In future work, we aim to investigate each component of SenSE in greater depth to develop more efficient and practical speech enhancement systems.

## C  THE USE OF LARGE LANGUAGE MODELS

In this work, large language models (LLMs) are employed exclusively for grammatical verification and language refinement. Specifically, LLMs are used to check the consistency of grammar, improve sentence fluency, and enhance overall readability of the manuscript. It is important to note that LLMs are not involved in any part of the algorithm design, data processing, model training, or experimental evaluation. Their role is limited to assisting in the preparation of the written text, ensuring that the presentation of this study meets the standards of academic writing.

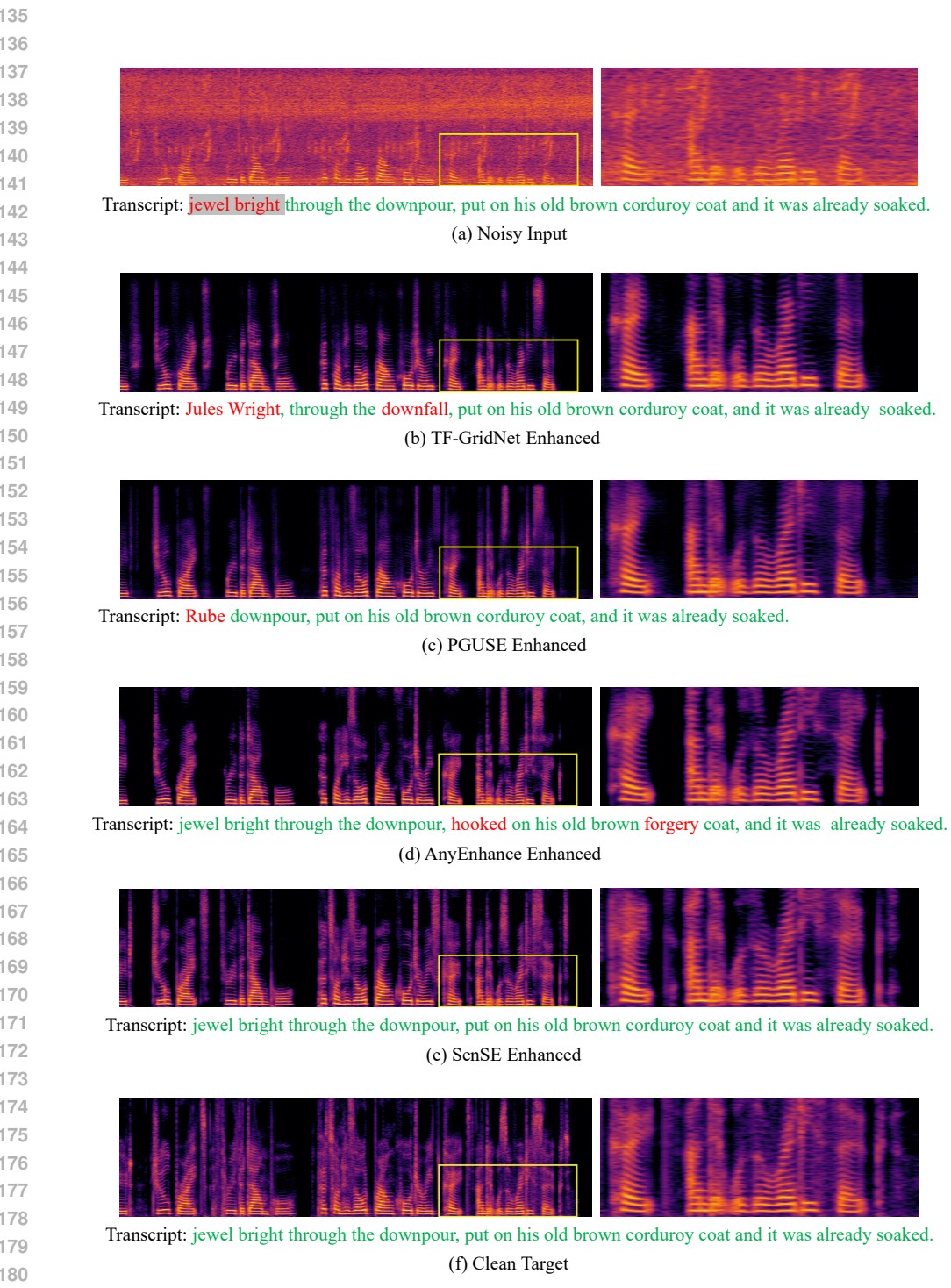

Figure 5: Spectrogram visualizations of the enhanced results from SenSE and selected baseline models. For each spectrogram, the corresponding transcription generated by the Whisper-large-v3 model is provided below, where green text denotes correct recognition, red text indicates errors, and gray background marks omissions.

