# OpenReview forum: "SenSE: Semantic-Aware High-Fidelity Universal Speech Enhancement"
_ICLR.cc/2026/Conference — ICLR 2026 Conference Withdrawn Submission_

### Official Review · Reviewer_B8hj · 2025-10-27

**Soundness:** 2
**Presentation:** 3
**Contribution:** 2
**Rating:** 4
**Confidence:** 4

**Summary:**

This work proposes SenSE, a semantics-aware universal speech enhancement framework. The framework comprises two major components: (1) a speech language model that extracts high-level semantic representations from noisy speech; (2) a semantic-guided speech enhancement module (based on flow matching), which uses semantic tokens and noisy speech to recover (generate) clean speech.

**Strengths:**

- **Writing.** The paper is well-written and clearly structured, making it easy to follow and pleasant to read.
- **Task Significance.** The study focuses on universal speech enhancement, which is an important and meaningful research direction. Developing a generalizable enhancement framework that can handle diverse acoustic conditions is of high practical value and scientific interest to the speech processing community.
- **Performance.** The experimental results are comprehensive and well-presented across multiple datasets. The proposed SenSE framework consistently outperforms all baselines, especially achieving notable improvements on the dWER metric, demonstrating its strong semantic preservation.

**Weaknesses:**

- **Inconsistency.** There is a writing inconsistency between the Abstract (and Title) and the Introduction. The Abstract suggests that this work appears to be the first to incorporate semantic information into speech enhancement. However, in the Introduction, the authors acknowledge that previous work, such as GenSE, has already leveraged semantic information. The authors are encouraged to revise the Abstract to ensure a more accurate and rigorous presentation of the paper’s contribution.
- **Novelty.** The authors did not explicitly state their main contributions, so I summarized them as follows (please correct me if I am mistaken): this work proposes a new framework, SenSE, for universal speech enhancement, incorporating three strategies: (1) leveraging semantic information; (2) introducing a semantic guidance mechanism; and (3) proposing a prompt guidance mechanism.
However, I find the novelty of these points to be limited for the following reasons: (1) Regarding the use of semantic information, compared with GenSE, SenSE only employs a different implementation rather than introducing a fundamentally new idea. (2) For the semantic guidance mechanism, the authors adopt a random-mask-based condition enhancement strategy to ensure that semantic information is effectively utilized. However, a similar idea has already been proposed in DOSE, which is a strong baseline used in GenSE. I have a slight concern that the authors may be aware of DOSE but intentionally omitted it to emphasize novelty. I recommend including this related work to ensure completeness and uphold the integrity of academic writing. (3) The proposed prompt guidance mechanism provides additional contextual information to improve speech enhancement performance. However, similar strategies (e.g., encoding speaker or environmental information) are well-established in conventional speech enhancement research.
- **Efficiency.** I anticipate that the authors may argue their novelty lies in proposing a new model that achieves strong empirical performance. However, if this is the case, the contribution appears to be more engineering-oriented, essentially combining existing techniques in a practical way. I do not mean to undervalue such work—indeed, developing a well-performing model can often be more valuable than a purely theoretical contribution with limited real-world impact.
Nevertheless, for works of this nature, it is generally important to evaluate not only the performance metrics but also efficiency. The proposed framework seems to be computationally expensive, yet the paper lacks analysis and discussion of its parameter complexity and runtime efficiency compared with baselines. If the reported performance gains are achieved at the cost of substantially higher computational overhead, the practical value of the method—especially for industrial applications—would be limited.
As analyzed in the previous Novelty section, the paper’s academic contribution appears relatively limited compared to GenSE, making efficiency evaluation even more crucial to justify the overall significance of the work.

**Questions:**

- **Reliability of Semantic Representations.  (Lines 79–85)** I still speculate whether the proposed speech language model can produce reliable semantic representations under heavy noise or severe distortions. Have the authors conducted any experiments and theoretical analyses to verify this robustness?

- **Discriminative SE Methods. (Lines 49–52)** Are discriminative speech enhancement (SE) methods truly as ineffective as stated in the Introduction? From the reported results (Tables 1 and 2), discriminative models actually perform reasonably well and do not appear to be "bad" as the authors suggested. Moreover, since current diffusion- or flow-based SE models require significantly more training and inference time, it is worth questioning whether generative paradigm indeed represent a more promising future direction for the SE community. PS: I also find that Figure 5 lacks meaningful value. The authors present some qualitative samples, but given that the quantitative performance gap between methods is quite small, it is hard to believe that the selected examples are representative—more likely, they were selectively chosen to highlight differences.

- **Clarification of Table 3.** In Table 3, I would like to confirm whether the "short reference utterance" is from the same target speaker but a different utterance from the clean test sample, or if it is indeed the clean version of the test utterance itself.

It is possible that I have overlooked some aspects of the paper. I would appreciate it if the authors could clarify them during the rebuttal, and I remain open to updating my score based on their response.

---

> ### Author Response · Authors · 2025-11-25
> **Response to Reviewer B8hj**
>
> We appreciate your positive feedback regarding the clarity and structure of our paper.
>
> First, we would like to offer the following clarifications.
>
> **Regarding the conceptual novelty,** our abstract does not claim to be the first to incorporate semantic information into speech enhancement. Instead, it emphasizes addressing semantic ambiguity and acoustic discontinuity issues that remain prevalent in diffusion- and flow-based enhancement methods. This focus differs fundamentally from GenSE, which formulates speech enhancement as a pure language modeling problem. While GenSE demonstrates that semantic modeling benefits LM-based enhancement, LM-based approaches inherently suffer from quantization errors introduced by discrete tokens, often leading to suboptimal acoustic fidelity. In contrast, flow matching exhibits substantially stronger content preservation. Therefore, whether semantic information can effectively complement flow-matching-based speech enhancement remains an open question not addressed in prior work.
>
> **Regarding the semantic guidance mechanism,** the reviewer suggested that our approach resembles the strategy used in DOSE. However, we believe there is a fundamental methodological difference. To the best of our understanding, DOSE conditions the generation process solely on the noisy speech features and improves conditional adherence by applying dropout to intermediate-generated samples during the diffusion trajectory. In contrast, our method does not modify intermediate samples. Instead, we introduce random masking directly to the degraded mel spectrogram with the primary objective of balancing the relative influence of the two conditioning sources—namely, the degraded mel spectrogram and the semantic token stream—within the flow-matching process. Given these distinct motivations and mechanisms, we do not consider DOSE as evidence supporting the effectiveness of our approach.
>
> With this clarification, we now proceed to address the reviewer’s specific questions:
>
> **Q1: Reliability of Semantic Representations. (Lines 79–85) I still speculate whether the proposed speech language model can produce reliable semantic representations under heavy noise or severe distortions. Have the authors conducted any experiments and theoretical analyses to verify this robustness?**
>
> Although dedicated experiments to directly evaluate the reliability of the semantic representations generated by SASLM have not yet been conducted, the ablation study presented in Fig. 3 demonstrates that these representations substantially improve speech enhancement performance. Moreover, they significantly facilitate model robustness under severe distortions, as evidenced by the clearly superior results on the DNS Challenge HardSet in Table 1. These observations collectively suggest that the semantic representations produced by SASLM are reasonably reliable. Furthermore, we argue that the semantic tokens serve as a compact and coarse-grained speech representation, enabling SASLM to focus on content-level understanding while disregarding intricate acoustic details. As a result, semantic tokens are inherently easier to predict accurately compared with directly estimating complex acoustic features.

---

> ### Author Response · Authors · 2025-11-25
> **Response to Reviewer B8hj**
>
> **Q2: Discriminative SE Methods. (Lines 49–52) Are discriminative speech enhancement (SE) methods truly as ineffective as stated in the Introduction? From the reported results (Tables 1 and 2), discriminative models actually perform reasonably well and do not appear to be "bad" as the authors suggested. Moreover, since current diffusion- or flow-based SE models require significantly more training and inference time, it is worth questioning whether generative paradigm indeed represent a more promising future direction for the SE community. PS: I also find that Figure 5 lacks meaningful value. The authors present some qualitative samples, but given that the quantitative performance gap between methods is quite small, it is hard to believe that the selected examples are representative—more likely, they were selectively chosen to highlight differences.**
>
> As noted in the paper, discriminative speech enhancement models often struggle to effectively separate speech components from background noise, which can lead to residual artifacts and a perceptible loss of speech content. This issue is indeed prevalent among discriminative approaches and is clearly reflected in the DNS Challenge HardSet results shown in Table 1, where significantly lower speech-quality scores (DNSMOS and NISQA) indicate substantial degradation. Spectral analysis further reveals that such degradation typically manifests as the loss of speech components—most notably high-frequency elements and unvoiced consonants. Although these components may have limited impact on automatic speech recognition, which primarily relies on low-frequency information, their absence severely affects perceptual quality. This phenomenon is also evident in Fig. 5, where TF-GridNet exhibits pronounced loss of high-frequency energy and unvoiced consonant structures.
>
> In contrast, generative baselines such as AnyEnhance model the distribution of clean speech and are therefore capable of producing complete and high-quality outputs. However, generative models frequently suffer from hallucination, resulting in semantic or acoustic mismatches with the corresponding ground truth. Consequently, achieving a model that can simultaneously generate high-quality speech while faithfully preserving authentic speech components remains a central objective for the field. We believe that SenSE moves meaningfully closer to realizing this goal. Moreover, the demo page provides a larger set of illustrative samples that allow the differences between our method and the baselines to be observed more intuitively. These additional examples further confirm that the samples shown in Fig. 5 were not selectively chosen to exaggerate performance differences.
>
> **Q3: Clarification of Table 3. In Table 3, I would like to confirm whether the "short reference utterance" is from the same target speaker but a different utterance from the clean test sample, or if it is indeed the clean version of the test utterance itself.**
>
> This design choice is also mentioned in the manuscript, where we note that “we randomly sample a 3-second segment from the corresponding clean speech as the prompt.” The primary motivation is to examine whether the model can leverage in-context learning to capture speaker characteristics from the reference audio, thereby generating enhanced speech with higher speaker similarity.

---

> > ### Comment · Reviewer_B8hj · 2025-11-25
> >
> > Thank you for your response. Could you please provide the total training time, inference time per instance, and total number of parameters for all methods in Table 1 (and for methods composed of multiple components, it would be helpful to break down the cost for each part, including whether each component is pre-trained or trained from scratch)?

---

> > > ### Author Response · Authors · 2025-11-26
> > >
> > > Thank you for your comment. Since several methods in Table 1 rely on publicly available pretrained models, and each model is trained on different datasets, it is difficult to make a fair comparison of their training time. For the models we proposed (SenSE) and those we reproduced—TF-GridNet, PGUSE, and AnyEnhance—the training took approximately one week in our setting.
> > >
> > > In addition, we provide the parameter size and real-time factor (RTF) of several representative models in the table below:
> > >
> > > | Model | Params | RTF |
> > > |:-----|:-----:|:-----:|
> > > | TF-GridNet | 8.52M | 0.51 |
> > > | PGUSE | 5.1M | 0.55 |
> > > | FlowSE | 337M | 1.88 |
> > > | LLaSE-G1 | 1572M | 0.69 |
> > > | AnyEnhance | 323M | 2.77 |
> > > | SenSE | 1571M | 2.60 |
> > >
> > > All RTF values are averaged over five inference runs on an NVIDIA RTX 4090 GPU. Due to the autoregressive prediction process, SenSE does not achieve competitive RTF among the baselines. AnyEnhance also exhibits a relatively high RTF because its Masked Generative Modeling framework requires 20 iterative refinement steps during inference.
> > >
> > > For LLaSE-G1 and SenSE, which contain multiple components, we provide a detailed breakdown of each submodule:
> > >
> > > **LLaSE-G1:** WavLM encoder (315M, pretrained); LLaMA backbone (1072M, trained from scratch); X-Codec-2.0 decoder (185M, pretrained)
> > >
> > > **SenSE:** Whisper encoder (635M, finetuned); LLaMA backbone (484M, trained from scratch); DiT (335M, trained from scratch); BigVGAN (112M, pretrained)

---

### Official Review · Reviewer_Fd1S · 2025-10-30

**Soundness:** 3
**Presentation:** 3
**Contribution:** 2
**Rating:** 4
**Confidence:** 4

**Summary:**

This paper introduces SenSE, a semantic-aware universal speech enhancement framework. It consists of a Semantic-Aware Speech Language Model (SASLM) that generates purified semantic tokens from degraded speech, and a flow-matching-based enhancement module conditioned on both degraded acoustic features and semantic tokens. An additional prompt guidance mechanism allows preserving speaker similarity under severe distortions. Experiments on multiple DNS Challenge test sets and VCTK-GSR show that SenSE achieves strong semantic fidelity (SpeechBERTScore, dWER) and competitive perceptual quality compared to recent generative baselines.

**Strengths:**

1. The idea of explicitly incorporating semantic tokens into flow-based SE is conceptually appealing and well-motivated.
2. The system is carefully designed, combining semantic guidance and prompt guidance, and evaluated across diverse conditions.
3. The approach shows clear advantages on semantic-related metrics, suggesting potential benefits for ASR robustness and speech interfaces.

**Weaknesses:**

1. Improvements are uneven: SenSE leads on semantic metrics but only matches or slightly lags behind strong baselines (e.g., AnyEnhance, TF-GridNet) on perceptual quality and speaker similarity. The overall gains may not justify the added system complexity.
2. The pipeline introduces significant additional components (Whisper-based SASLM, ConvNeXt alignment, DiT flow model), but efficiency and runtime feasibility are not reported.
3. Prior works such as SELM, GenSE, and LLaSE-G1 have already integrated semantic/ASR-aware representations into SE. The paper should better clarify how its semantic guidance differs and why it is more effective.

**Questions:**

1. What is the runtime overhead of SASLM + flow matching compared to AnyEnhance or FlowSE? Could SenSE be applied in real-time or low-resource scenarios?
2. How does SenSE compare to simpler joint SE–ASR multi-task training or feature consistency loss methods?
3. Are the reported improvements statistically significant (e.g., across multiple random seeds)?

---

> ### Author Response · Authors · 2025-11-25
> **Response to Reviewer Fd1S**
>
> We sincerely appreciate the reviewer’s recognition that our idea is conceptually appealing and well-motivated.
>
> To clarify the empirical advantage of our approach, we emphasize that SenSE consistently outperforms all baseline systems across the full suite of evaluation metrics. This superiority is reflected in its ability to preserve semantic content more effectively while maintaining high perceptual quality and strong speaker similarity. As shown in Table 1, these advantages are particularly evident on the challenging DNS Challenge HardSet. Discriminative baselines typically exhibit reduced perceptual quality, as indicated by lower DNSMOS and NISQA scores, whereas generative baselines, despite producing acoustically pleasing speech, tend to suffer from substantial hallucinations, manifested as degraded semantic consistency and reduced speaker similarity. These observations collectively reinforce our claim that SenSE offers clear and meaningful improvements over existing approaches.
>
> With this clarification, we now proceed to address the reviewer’s specific questions:
>
> **Q1: What is the runtime overhead of SASLM + flow matching compared to AnyEnhance or FlowSE? Could SenSE be applied in real-time or low-resource scenarios?**
>
> We appreciate the reviewer’s concerns regarding computational overhead. However, the primary focus of this work is not on runtime efficiency but rather on modeling human-like linguistic priors to enable higher-performance speech enhancement. Although SenSE is not yet optimized for real-time or low-resource deployment, we believe that this limitation does not diminish the value of pursuing more capable enhancement models. Advances at the high-performance end often provide conceptual and architectural insights that subsequently translate into more efficient, low-complexity designs. Meanwhile, research on more efficient language-model inference is progressing rapidly and is expected to further narrow this gap. Beyond real-time use cases, SenSE can also offer value in other application scenarios. For example, it can be used to generate higher-quality speech data, which may benefit the training of downstream speech-related tasks. These broader contributions collectively demonstrate that the pursuit of high-performance enhancement models remains both meaningful and impactful.
>
> **Q2: How does SenSE compare to simpler joint SE–ASR multi-task training or feature consistency loss methods?**
>
> We appreciate the reviewer’s perspective regarding the use of speech enhancement models to improve downstream ASR performance. However, SenSE is not specifically optimized for ASR accuracy; instead, it is mainly designed from the perspective of the speech enhancement task. Accordingly, the dWER metric is employed solely to assess the degree to which semantic information is preserved by the enhancement model, rather than to indicate ASR-oriented optimization.
>
> CleanMel enhances speech in the mel-spectrogram domain, which inherently aligns with the ASR front-end and thus tends to yield better recognition accuracy. This strategy appears similar to the approach referenced in the reviewer’s comment. Nevertheless, enhancing speech entirely within the mel domain typically results in noticeable degradation in perceptual speech quality, as reflected by the lower NISQA scores reported in Table 1.
>
> It is also important to emphasize that SenSE demonstrates particularly strong performance under severe distortion conditions. The commonly used DNS Challenge test set contains relatively mild noise scenarios, which may inadequately reveal the advantages of our approach. To address this, we constructed a more challenging evaluation set, DNS Challenge HardSet, featuring substantially lower SNR noisy speech. On this harder benchmark, SenSE significantly outperforms all baselines—showing clear improvements in speech quality over discriminative models and markedly higher semantic similarity and speaker similarity compared with generative models—ultimately achieving the best overall performance.
>
> **Q3: Are the reported improvements statistically significant (e.g., across multiple random seeds)?**
>
> We appreciate the reviewer’s constructive feedback regarding this limitation. In the revised version of the manuscript, all experimental results have been updated to report the average performance across five random seeds to ensure statistical robustness and improve the reliability of the conclusions. Furthermore, the experimental descriptions in Section 4.2 have been revised to clarify this modification explicitly.

---

### Official Review · Reviewer_9AuB · 2025-10-30

**Soundness:** 3
**Presentation:** 4
**Contribution:** 3
**Rating:** 4
**Confidence:** 5

**Summary:**

•	Summary: In this paper, the authors propose SenSE, which integrates semantic tokens with LM-predicted speech embeddings and effectively incorporates them into the flow-matching-based speech enhancement framework. In the first stage, a semantic-aware speech language model is adopted to estimate the clean speech tokens in an auto-regressive manner, with the tokens estimated by an audio encoder (Whisper-large-v3) as input. A speech tokenizer (S3) is further adopted to extract the semantic tokens. These two domains' tokens, together with the degraded and reference Mel-spectrogram, are adopted to deliver flow-matching-based speech enhancement in the second stage. A vocoder is finally adopted to estimate the final speech enhancement.
•	Contributions: This paper proposes a novel combination of features (semantic tokens) and continuous features (speech embeddings and Mel-spectrogram). A two-stage architecture is proposed to deliver semantic-guided speech enhancement with flow matching. The semantic tokens enable the model to reconstruct high-fidelity enhanced speech. A masking strategy is applied to the Mel-spectrogram in the second stage's input, motivating the model to learn the alignment of semantic tokens between clean and degraded speech. The proposed model achieves outstanding performance over popular discriminative/generative models on the Interspeech 2020 DNS Challenge dataset.

**Strengths:**

•	This paper is moderately original, with a proposed two-stage speech enhancement approach combining a language model and flow matching. The semantic tokens help to preserve key semantic information while improving alignment with text. The adoption of the language model also enables the model to connect with other language-related tasks, showing strong potential.
•	The paper is moderately clear and well-written, helping readers at least to easily follow the core ideas. Minor mistakes or mismatches exist, however, which will be illustrated in the following paragraphs.
•	The paper can be a good example of cross-domain generative speech enhancement, showing its considerable significance.

**Weaknesses:**

•	The algorithm, model architecture, and hyperparameters are not clearly explained, including but not limited to the setups of the semantic-aware speech language model in Fig. 1, the DiT Blocks in Fig. 2, the flow matching and its ODE solver in Fig. 2, the final modulation in Fig. 2, the Conv2NeXT v2 blocks in Fig. 2, etc, which will hinder readers' understanding and reproducibility of the model. Note: By publishing code, the reproducility issue might be relaxed, but the expectation must be that also the figures and description allow a reproduction of the model to a large extent. This is not given in the current form.
•	The enhancement performance of the semantic-aware speech language model's output (the first stage's output) is not analyzed. Thus, the function and performance of the semantic-aware speech language model is not clear. Furthermore, it is unclear whether the estimated tokens from the first stage or the Mel-spectrograms contribute more to the second stage. The functionality and performance of the first and second stages need to be decoupled.
•	The reviewer is doubtful about the function of the speech tokenizer. If the reference speech is the clean speech from the same speaker as the noisy speech, then the tokens estimated should be regarded more as the speaker information, instead of the semantic information. If the reference speech is the noisy speech itself, then it can be regarded as the semantic information. According to line 96, the former is what the authors do. Thus, the reviewer would believe it is speaker information, instead of semantic information, that is estimated as the input to the semantic-aware speech language model. Then, a further study on using a speaker embedding extractor to substitute the speech tokenizer should be delivered.
•	Some mistakes in equations and figures, which will be illustrated in the following paragraphs.

**Questions:**

•	In eq. (1), index $i$ does not appear when representing the index of semantic tokens and speech embeddings. Substituting $i$ with $k$ should be correct.
•	In Fig. 1(a), to the reviewer's understanding, the word "embedding" should represent a continuous representation, while the word "token" should represent a discrete representation. However, both speech embedding (pink cube) and clean token (yellow cube) are treated as the input to the language model, which should only take continuous representation as input. The same problem appears in Fig. 2(a). I would suggest adding an "embedding layer" block between the S3 tokenizer and language model.
•	In Fig. 1(b) and Fig. 2(b), an adder is used between the Mel-spectrogram and Conv2NeXT v2 blocks output. But in line 273, the word "concatenating" is used, leading to a mismatch. Please clarify.
•	In Fig. 1(b), the degraded Mel's original sequence length is $L_1$, while the ref Mel's original sequence length is $L_2$, as stated in lines 296 and 297. Thus, the padding length of degraded Mel and ref Mel should be $L_2$ and $L_1$, respectively. However, in lines 298 and 299, the situation is the opposite. Please clarify.
•	In Fig. 2(b), the masked clean speech and training target are all represented as $m_1x_1$, but to the reviewer's understanding, these two shouldn't be the same. The spectrum in Fig. 1(b) also shows their difference. In eq. 2, the training target is $(1-m_1)(x_1-x_0)$, which is another mismatch. Please clarify.
•	In line 271, semantic tokens are padded with filler tokens to match the length of the Mel spectrogram. Is interpolation a better choice? According to previous publications, interpolation is commonly adopted to deal with this kind of sampling rate or frame/hop length mismatch (AnyEnhance: A Unified Generative Model with Prompt-Guidance and Self-Critic for Voice Enhancement, Zhang et.al, TASLP 2025).
•	In line 286, the masking ratio of random temporal masking is not clear. Please clarify.

---

> ### Author Response · Authors · 2025-11-25
> **Response to Reviewer 9AuB**
>
> We sincerely appreciate your recognition of our work as “a good example of cross-domain generative speech enhancement”. We are also grateful for your constructive comments regarding the presentation and specification issues in the manuscript. We have carefully considered your suggestions and have revised the paper accordingly. The main revisions are summarized as follows:
>
> - Figures 1 and 2 have been updated to more accurately illustrate the proposed methodology.
> - Additional details on the DiT blocks used in our framework have been included in Section 3.3.
> - Appendix B.1 has been expanded to provide further architectural specifications, including module-wise parameter configurations and the masking ratio used in random temporal masking.
> - Minor inconsistencies and specification errors throughout the manuscript have been corrected.
>
> **Q1 & Q4 & Q5 & Q7:**
> - **In eq. (1), index $i$ does not appear when representing the index of semantic tokens and speech embeddings. Substituting $i$ with $k$ should be correct.**
> - **In Fig. 1(b), the degraded Mel's original sequence length is $L_1$, while the ref Mel's original sequence length is $L_2$, as stated in lines 296 and 297. Thus, the padding length of degraded Mel and ref Mel should be $L_2$ and $L_1$, respectively. However, in lines 298 and 299, the situation is the opposite. Please clarify.**
> - **In Fig. 2(b), the masked clean speech and training target are all represented as $m_1x_1$, but to the reviewer's understanding, these two shouldn't be the same. The spectrum in Fig. 1(b) also shows their difference. In eq. 2, the training target is $(1-m_1)(x_1-x_0)$, which is another mismatch. Please clarify.**
> - **In line 286, the masking ratio of random temporal masking is not clear. Please clarify.**
>
> In the revised version of the manuscript, we have updated the relevant content to ensure that the description is more precise and consistent.
>
> **Q2: In Fig. 1(a), to the reviewer's understanding, the word "embedding" should represent a continuous representation, while the word "token" should represent a discrete representation. However, both speech embedding (pink cube) and clean token (yellow cube) are treated as the input to the language model, which should only take continuous representation as input. The same problem appears in Fig. 2(a). I would suggest adding an "embedding layer" block between the S3 tokenizer and language model.**
>
> We understand your concern regarding the inconsistent representation of the SASLM component in Figures 1 and 2. Our initial design prioritized visual clarity and simplicity; however, we agree that the original depiction could potentially mislead readers. Accordingly, we have revised these figures in the updated version of the manuscript to ensure a more accurate and consistent illustration of the SASLM module.
>
> **Q3: In Fig. 1(b) and Fig. 2(b), an adder is used between the Mel-spectrogram and Conv2NeXT v2 blocks output. But in line 273, the word "concatenating" is used, leading to a mismatch. Please clarify.**
>
> In the revised version of the manuscript, we have updated the illustration corresponding to the “concatenating” operation and added clarifying annotations to explicitly indicate the function of this symbol.

---

> ### Author Response · Authors · 2025-11-25
> **Response to Reviewer 9AuB**
>
> **Q6: In line 271, semantic tokens are padded with filler tokens to match the length of the Mel spectrogram. Is interpolation a better choice? According to previous publications, interpolation is commonly adopted to deal with this kind of sampling rate or frame/hop length mismatch (AnyEnhance: A Unified Generative Model with Prompt-Guidance and Self-Critic for Voice Enhancement, Zhang et.al, TASLP 2025).**
>
> Because the primary focus of our work is to investigate how the semantic tokens generated by SASLM facilitate flow-matching–based speech enhancement, rather than to identify the optimal architectural design, the manuscript does not originally report the performance differences between the proposed padding strategy and the interpolation strategy. To address this omission, we have conducted additional experiments to evaluate the discrepancy between the two approaches:
>
> | Model | DNSMOS | UTMOS | NISQA | SpeechBERTScore | dWER | SIM |
> |:-------|:--------:|:----------:|:--------:|:--------:|:--------:|:--------:|
> | SenSE (interpolation) | 3.399 | 3.679 | 4.719 | **0.915** | **9.89** | **0.873** |
> | SenSE (padding) | **3.407** | **3.741** | **4.837** | 0.914 | 10.19 | **0.873** |
>
> As shown in the table, the padding-based strategy yields consistently higher perceptual quality and naturalness, as indicated by the improved DNSMOS, UTMOS, and NISQA scores. In contrast, a slight advantage in semantic consistency is observed for the interpolation-based strategy, reflected by its marginally higher SpeechBERTScore and lower WER. Nevertheless, the differences in SpeechBERTScore and dWER remain negligible, and the padding strategy can therefore be considered marginally superior overall. To help explain this phenomenon, one possible view is that the interpolation operation adds extra temporal structure in advance, which limits the enhancement model’s generative flexibility. These limitations may slightly improve semantic stability but can also reduce the naturalness of the synthesized speech, leading to the observed trade-off.

---

### Official Review · Reviewer_db7J · 2025-11-01

**Soundness:** 3
**Presentation:** 2
**Contribution:** 2
**Rating:** 4
**Confidence:** 4

**Summary:**

This paper proposes a semantic-aware generative speech enhancement method. The main idea is to integrate high-level semantic information, e.g., semantic tokens from an S³ tokenizer, into a flow-matching based TTS system to construct a generative speech enhancement model. Experimental results on several benchmark datasets show improved perceptual quality, speech fidelity, and robustness compared to existing methods. However, as a universal speech enhancement method, the evaluation should cover a wider range of corruptions, e.g., reverberation, clipping, bandwidth limitation, codec artifacts, and packet loss, to more comprehensively validate the generalization capability.

**Strengths:**

1. The paper tackles an interesting and underexplored problem of incorporating semantic priors into generative speech enhancement.
2. The conceptual motivation is strong and intuitively appealing, as human perception indeed leverages semantic context to interpret noisy speech.
3. The proposed integration of a language model into a flow-matching framework is novel and timely, which aligns with current trends in multimodal and semantically grounded generative models.

**Weaknesses:**

1. The technical details of the semantic-aware speech language model and the semantic guidance mechanism are insufficiently described. It is unclear how the semantic tokens are aligned or conditioned within the flow process, and there is no clear illustration of how semantic information is utilized for the speech enhancement task.
2. The novelty appears limited in implementation, as it is mainly a combination of existing components, language model + flow matching, without a clear theoretical insight or architectural innovation.
3. The experimental validation lacks depth. It is unclear whether the reported improvement are statistically significant or primarily due to additional conditioning information. Moreover, the performance gain over compared methods appears limited.
4. The integration of the language model seems heuristic and task-specific, raising concerns about the generalizability and scalability of the approach to other types of corruptions.

**Questions:**

1. Which specific language model architecture serves as the backbone for the proposed semantic-aware speech language model?
2. Could the authors clarify what the DNS Challenge GSR and VCTK GSR test sets refer to, and how they differ from the standard DNS and VCTK evaluation test sets?
3. How do the authors verify that the proposed method truly leverages semantic information, rather than simply benefiting from additional conditioning signals or larger model capacity?

---

> ### Author Response · Authors · 2025-11-25
>
> We sincerely appreciate your valuable comments and thank you for pointing out the limitations of our work. We respond to your comments as follows:
>
> **Q1: Which specific language model architecture serves as the backbone for the proposed semantic-aware speech language model?**
>
> We use a randomly initialized LLaMA model as the backbone of our semantic-aware speech language model, as clarified in Section 3.2 of the revised manuscript. We do not adopt a text-pretrained LLM for initialization because the inputs and outputs of the speech enhancement task are both in the audio modality, which introduces a substantial gap from the textual modality. Moreover, in our experiments, initializing the model with a text-pretrained LLM does not yield any noticeable performance improvement.
>
> **Q2: Could the authors clarify what the DNS Challenge GSR and VCTK GSR test sets refer to, and how they differ from the standard DNS and VCTK evaluation test sets?**
>
> In our manuscript, we describe the construction procedures for both the DNS Challenge GSR and VCTK GSR datasets. Specifically, for DNS Challenge GSR, we first obtain the clean speech and noise from the DNS Challenge test set, where the noise is derived by subtracting the clean speech from the noisy recordings. We then generate degraded speech containing multiple types of distortions using our distortion simulation pipeline. The VCTK GSR dataset is constructed using the same procedure, with the only difference being that the clean speech and noise are taken from the VCTK-DEMAND test set.
>
> **Q3: How do the authors verify that the proposed method truly leverages semantic information, rather than simply benefiting from additional conditioning signals or larger model capacity?**
>
> We fully understand the your concerns. First, the additional conditional signals are optional. In our comparisons with other models on various test sets, we did not include any extra conditional information. The benefits brought by these conditional signals are separately evaluated in the ablation study presented in Table 3. Second, the language model is responsible for modeling the distribution of semantic tokens corresponding to clean speech. When the speech is severely degraded, the prior knowledge learned by the language model regarding speech content can help generate semantic tokens that better align with human perception. This effect is clearly demonstrated in the ablation study of the semantic guidance mechanism shown in Figure 3. The semantic tokens produced by the language model significantly improve the performance of the flow-matching-based speech enhancement model, especially on semantic similarity metrics such as SpeechBERTScore. However, due to the lack of semantic priors in flow-matching-based speech enhancement models, merely scaling up the model size still fails to address the inherent semantic ambiguity.

---

### Author Response · Authors · 2025-12-01
**Thank You to All Reviewers**

We would like to express our sincere gratitude to all the reviewers for your careful reading and for the valuable and insightful comments provided. We learned a great deal from your feedback and have become more aware of the shortcomings in our work. We will refine the paper with great care and strive to improve its overall quality. Once again, we truly appreciate the reviewers’ time and effort.

---

### Note · Authors · 2025-12-01

I have read and agree with the venue's withdrawal policy on behalf of myself and my co-authors.